# Laser Structuring and DLC Coating of Elastomers for High Performance Applications

**DOI:** 10.3390/ma15093271

**Published:** 2022-05-02

**Authors:** Sönke Vogel, Andreas Brenner, Bernadette Schlüter, Bernhard Blug, Franziska Kirsch, Tamara van Roo

**Affiliations:** 1Chair for Laser Technology LLT, RWTH Aachen University, 52074 Aachen, Germany; 2Fraunhofer Institute for Laser Technology, 52074 Aachen, Germany; andreas.brenner@ilt.fraunhofer.de; 3Fraunhofer Institute for Mechanics of Materials, 79108 Freiburg, Germany; bernadette.schlueter@iwm.fraunhofer.de (B.S.); bernhard.blug@iwm.fraunhofer.de (B.B.); 4Fraunhofer Institute for Structural Durability and System Reliability LBF, 64289 Darmstadt, Germany; franziska.kirsch@lbf.fraunhofer.de (F.K.); tamara.van.roo@lbf.fraunhofer.de (T.v.R.)

**Keywords:** diamond like carbon, DLC, tribology, rubber, flexible, nature inspired, coating, friction, sealing, delamination

## Abstract

Even though hard, low friction coatings such as diamond like carbon (DLC) would be beneficial for the performance and longevity of rubber seals, a crucial challenge remains. The elastic mismatch of rubber substrate and DLC coating prevents a fracture free coating application. In this work, a nature inspired approach is applied to render the stiff coating flexible and resilient to delamination at the same time by direct patterning. Rubber substrates were laser structured with tile patterns and subsequently DLC coated. Tensile and tribology tests were performed on structured and unstructured samples. Unstructured DLC coatings showed a crack pattern induced by the coating process, which was further fragmented by tensile stress. Coatings with tile patterns did not experience a further fragmentation under load. During continuous tribological loading, less heterogenous damage is produced for tile structured samples. The findings are ascribed to the relief of induced coating stress by the tile structure, meaning a more resilient coating.

## 1. Introduction

Rubber seals are everywhere and, in every dynamic seal, the rubber is responsible for 50–70% of friction losses. A hard, protecting, and low friction coating such as diamond like carbon (DLC) would significantly enhance the performance of dynamic rubber seals, as the adhesive forces between the rubber and its counterpart are reduced by interlaying a DLC film. Moreover, the viscoelastic contribution is reduced, though the former has been found to be more prominent [1,2]. Besides the already mentioned properties of DLC, good adhesion and flexibility are mandatory requirements. Whereas the coefficient of friction (CoF) reduction in DLC coatings compared to their pristine substrate has been shown [3,4,5,6,7,8,9], the latter properties are subject to ongoing research. In particular, the microstructure of the DLC film has been attributed to adhesion and flexibility performance [1,8,9,10]. Inherent to the coating process, resulting from a thermal coefficient mismatch of substrate and coating, is the formation of buckling cracks in the stiffer DLC. A dense crack pattern with small patches was found to enhance the adhesion [8] and flexibility [11]. The microstructure was related to substrate hardness [6] and surface chemistry and roughness [8,12] as well as coating conditions with preceding plasma treatment [1,2,7,9,10]. The latter method aims to passively influence the microstructure and is thus regarded as a subtype of microstructuring. Direct microstructuring was performed with a net mask. However, the tested substrate was “soft” aluminum and the smallest achievable patch size was 1 mm with 100 µm spacing [13], whereas patch sizes below 60 µm were found to unify good adhesion and CoF [1,2,9].

The primordial crack pattern of native coatings is formed by buckling and is crucial for attaining a flexible film. Under thermally induced compression, the coating either forms a blister and finally cracks at the blister ridge [14,15] or it wrinkles in a complex sinusoidal pattern and cracks at the trench [16,17]. The latter was found for DLC coatings on compliant substrates [18] and is unavoidable in natively grown films. However, the film is prone to crack further under load and debris is generated [1,7,8,12].

In this study, we aim to control the patch size—which has been identified as the most influential parameter—by direct ultrashort pulse (USP) laser microstructuring of the substrate. Generally, the laser ablation of polymers and elastomers is similar to dielectrics [19]. Electrons are initially excited from the valence band to the conduction band by single photon, multiphoton or strong field excitation [20]. A further excitation by inverse bremsstrahlung starts an avalanche ionization. Subsequently, the lattice is thermalized by electron–phonon coupling [21,22]. For USP lasers, in contrast to nanosecond lasers, the heating is isochoric. The adiabatic cooling of the isochoric material causes (a) direct vaporization, (b) fragmentation due to supercritical relaxation, (c) phase separation and material ejection, or (d) phase transition without ablation [23]. In the case of elastomers and polymers, photodecomposition or depolymerization can take place as well [24]. For longer pulse durations (i.e., nanoseconds), two ablation thresholds are distinguishable, attributed to photothermal and photochemical ablation [25]. Higher fluences are attributed to main chain scission [26,27,28] and lower fluences to thermal activation [25,29] or crosslinking [26]. Below threshold swelling [30,31,32] is a further indicator for chemical modification. For femtosecond irradiation, thermal effects are reduced and ablation is induced by main chain scission with reduced chemical modifications [27,33,34,35,36].

Whereas the native microstructure of DLC coatings on elastomer substrates is only indirectly controllable and still strongly dependent on many influential parameters, a direct patterning could be beneficial in several ways: First, the tribological properties of the film could be further enhanced by microstructures. Second, the microstructures could act as a lubricant reservoir. Third, the adhesion and flexibility could be improved further in comparison to passively controlled native films. The approach is inspired by scale armors found in nature, which are resilient and yet flexible.

We hypothesize that the proposed tile pattern is believed to surpass passively controlled films with regard to delamination performance. The tile pattern is expected to suppress wrinkling and fragmentation during loading, resulting in smoother, long term stable coatings. We did not aim to optimize the dimensions of the tile pattern to obtain a lower CoF. This study focuses on the long term adhesion of the coating which is reflected by a smaller change in the CoF compared to an unstructured sample.

## 2. Materials and Methods

### 2.1. Materials

Nitrile–butadiene–rubber (NBR) and fluorelastomers (FKMs) are the most common materials for sealing applications. To address as many sealing applications as possible, both NBR and FKM with different hardnesses (Shore A) are investigated. Furthermore, two different kinds of styrene-ethylene-butylene-styrene (SEBS) are examined as an example of thermoplastic elastomers. Table 1 shows the materials used and their Shore A hardness, measured similar to ISO 868 (digi test II, Barheiss Prüfgerätebau GmbH). Moreover, the oil and soot content was thermogravimetrically measured (TGA2, Mettler Toledo). The furnace was heated at 10 K/min and flushed with nitrogen. From 600 °C to 1000 °C, the flush gas was changed to air.

### 2.2. Laser Processing

The rubbers were processed with ultrashort laser pulses (USPs) utilizing a PHAROS (LightConversion, Lithuania) system with a repetition rate of 200 kHz, 200 fs pulse width at a central wavelength of 343 nm. Three different surface topographies were engraved:Tile pattern with a hatch distance of 60 µm and a trench width and depth of 10 µm and 30 µm, respectively. In the further text, this is referred to as S1.Tile pattern with a hatch distance of 30 µm and a trench width and depth of 10 µm (S2).Single depressions distanced by 20 µm in two perpendicular directions with a diameter and depth of 10 µm (S3).

All trenches were processed with a spatial pulse overlap of 75% and all depressions by a spatial overlap of 100% and 50 pulses. The trench depth and width were verified by cross sections, cut with a sharp razor blade. All microscope investigations utilized a Keyence VHX5000.

### 2.3. DLC (a-C:H) Coating of Rubber Substrates

The samples were cleaned in an ultrasonic bath with ethanol. The samples were then DLC coated in a standard capacitively coupled plasma enhanced chemical vapor deposition process (CCP PE-CVD) at a frequency of 13.56 MHz. The process started with a 5 min plasma cleaning in argon (35 SCCM) followed by a 2 min adhesion layer coating with tetramethylsilane (TMS, 17 SCCM) as precursor. The main coating gas (toluene vapor) was then ramped for 5 min into the gas mixture by substituting the TMS with toluene. The power in the process was held constant at 200 W and the end pressure reached in the toluene process was 2.3 Pa. The thickness was evaluated by Swanepoel’s method (Figure A1) [37] from optical spectra taken from 1000 nm to 2400 nm (Lambda 1050, Perkin Elmer, Waltham, MA, USA).

### 2.4. Tensile Tests

All quasistatic tensile tests were executed on a servohydraulic Zwick/Roell testing machine equipped with a 20 kN load cell at constant velocity of 6 mm/min. The tensile tests were performed on unstructured, uncoated as well as coated FKM Viton 75 samples (dimensions of the sample in Figure 1b). Magnified images (×100) of the coated surface were taken for different strains during the first elongation. The strain information was evaluated by optical gray scale correlation (VIC2D, Correlated Solutions^®^, Irmo, SC, USA).

The spatial frequency distribution (SPD) was obtained from the region in the range of |45°| along the elongation axis for NBR-H-AT-68 (see Appendix A for further explanations). It was assumed that crack formation due to tensile stress in the DLC coating is oriented preferentially perpendicular to the elongation direction. Therefore, a shift of the SPD to shorter wavelengths λ is associated with a higher crack density. In contrast, a shift to longer wavelengths corresponds to an expansion of existing cracks. The images were taken with a Keyence VHX-2000.

### 2.5. Tribology and Surface Degradation

Tribology tests were performed using a custom made tribometer with a ring on disc geometry and a contact area of 169.65 mm^2^. In order to test the elastomers, structured and DLC coated elastomer rings (NBR-H-AT-68, Ø_inner_: 9.2 mm; Ø_outer_: 21 mm, thickness: 2 mm) were glued on metal discs (AS_0821, 100Cr6) with cyanoacrylate glue and dried for one night (Figure 1a). As counterpart axial bearings, steel discs (AS_1528, 100Cr6, R_a_ 0.1 µm, R_Z_ 0.8 µm, RPC 64/cm, technical roughness) were used. A load of 0.3 MPa at 148 rpm (0.14 m/s) was applied for long term experiments under dry conditions, which lasted for 12 h. The structural development of the abraded surface area was captured using a Keyence VK 9700K laser scanning microscope with a step size in the *z*-direction of 0.5 µm at ×20 magnification.

## 3. Results

### 3.1. Laser Processing

All materials, independent of fillers or transparency, could be processed homogenously and consistently.

A cross section of NBR and a translucent SEBS are exemplarily depicted in Figure 2. Both show a similar dependency on the number of scans as indicated by the similar single shot threshold fluence F_Th_ (Table 1). High aspect ratio trenches (16:1) with a constant width of approximately 10 µm could be manufactured in all materials (Figure 2g).

### 3.2. DLC Coating of Rubber Substrates

The DLC coating of all rubber substrates was successful and no delamination or excessive substrate warping was detected. The DLC coating had an average thickness of 2.5 µm and a Vickers microhardness of 1350 µm. By SEM analysis, a continuous DLC coating was identified inside the trenches and depressions (Figure 3).

### 3.3. Tensile Test

Figure 4 shows the average (*n* = 3) stress–strain curves for uncoated and DLC coated FKM Viton 75, both unstructured and measured with a sample size. The elastic modulus of the uncoated sample (19.1 MPa) is lower than the elastic modulus of the DLC coated sample (28.9 MPa). Furthermore, the coating leads to higher tensile strength and higher elongation at break.

The DLC coating of the unstructured reference sample showed an irregular pattern of cracks with no recognizable preferred direction before elongation. According to the Fourier transformation, the wavelength distribution peaks around 90 µm and 100 µm parallel and perpendicular to the elongation direction, respectively. The DLC coating of the sample with structure S1 shows a regular grid pattern with 60 µm spatial separation in and perpendicular to the elongation direction. Small voids and substructures are recognizable in each slab. A similar coating morphology is found for structure S2 with DLC coating. However, some slabs are connected by a continuous DLC coating without a recognizable trench in between. Neither sample showed delamination of the DLC coating.

Reference: At 10% strain of the reference sample, a noticeable increase in cracks is identified visually, accompanied by the onset of the SPD shift towards shorter wavelengths (Figure 5). Until 40% strain, an ongoing SPD shift is observed. Further elongation shifts the SPD towards longer wavelengths. In real space, further fragmentation is not observed. However, the already existing cracks are expanded. The SPD shift asymptotically approaches the measured elongation, highlighted by the dotted line. After unloading, the cracks are completely closed and the SPD corresponds to the SPD before the tensile test.

Structure S1: Up to 34% strain, the existing trenches in the elongation direction are expanded while the trenches perpendicular to the elongation direction contract (Figure 6). They become fully closed at around 34% strain. Towards higher strains, different distortion phenomena of the slabs are observed: Pincushion distortions and meandering perpendicular to the elongation direction. Cracks seem to form inside some individual slabs. The SPD is constantly shifted towards longer wavelengths. The shift follows the elongation as highlighted by the dotted line.

Structure S2: An inspection of the DCL coating at 6% strain shows an advanced expansion of some trenches in contrast to others. Further straining intensifies this irregularity which ultimately leads to meandering trenches perpendicular to the elongation direction. The SPD peak shifts constantly towards longer wavelengths, following the elongation. Compared to S1, the SPD is broader, indicating an irregular expansion. For some samples, the DLC coating bridged the trenches, except the trench intersections, leading to a morphology as in S3 and a similar fracture behavior as the reference sample.

### 3.4. Tribology and Surface Degradation

The structures S1 to S3 as well as one unstructured reference sample were investigated, each with and without DLC coating. As expected, lower friction values at the end of the test of approx. 20–30% were obtained in the tribological experiments when the elastomer is coated with DLC (Figure 7a). This is independent of the structuring of the elastomer. Wear also decreased significantly due to the DLC coating. Figure 8 gives a comparison of the different structured and coated elastomer rings after the tribological experiments. The structured samples without DLC show significantly more macroscopic abrasion patterns after the tribological experiment than the unstructured sample. The DLC coating results in a different running in behavior of the specimens: Whereas for uncoated samples a decreasing CoF is typically observed, the coated samples exhibit a continuously increasing CoF. The initial high friction (µ = 0.6) occurring for the elastomer–steel contact (high adhesive contact) decreases to a run in condition (µ ~ 0.35) after about 1–2 h. The friction values of the DLC coated elastomers start at about 0.1 and—depending on the structure—develop quickly or slowly into a steadily increasing CoF, which reaches about µ = 0.2 after 12 h. For the DLC coated sample (+DLC), a very fast increase in the CoF within the first 30 min was found which then continued to increase steadily at a significantly lower rate.

The CoF of the S3+DLC specimen also exhibits a very strong increase within the first 20 min. In order to facilitate the comparison of the development of the friction coefficient of the DLC coated samples, the curves were normalized to the friction coefficient at the end of the test and plotted with a logarithmic time scale (Figure 2b). Both the unstructured and the point structured samples (S3) show two regimes for the CoF: An initial increase and then a moderate increase as compared to the latter. The two tile structures S1 and S2 do not show this intensive initial increase.

In the two top rows in Figure 8, laser scanning microscopic pictures (×20 magnification) of the DLC coated surfaces before and after the tribological experiment are compared. The unstructured DLC coated surface initially shows a complex structured morphology. As a result of the tribological loading, this structure becomes more fragmented. The patterns of the structured elastomer samples remain clearly visible when coated with DLC. A fragmentation of the DLC coating in the wear track (similar as for the unstructured specimen) for S3+DLC is equally noticeable. The other two DLC coatings (tile structures S1 and S2) do not show any clear change in the coating pattern.

## 4. Discussion

From the coating texture alone (Figure 6), the conclusion can be drawn that the trenches facilitate stress relief inherited from the coating process. During tensile tests, the individual tiles of structure S1 and S2 experienced no further fragmentation. The induced stress is completely relieved by the expansion of the trenches. The same principal is observed for the native, unstructured coating. However, a threshold crack density needs to be surpassed. This crack density is dependent on the film adhesion, tensile strength, and film thickness [38]. Up to 8% strain, preexisting cracks are expanded. Further straining fragments the film up to the threshold crack density at 35% strain. These findings are in accordance with [18]. However, one load cycle does not change the SPD of the reference sample when unloaded to 0% strain, which may indicate an insufficient resolution of the utilized optical system. Thus, the cracks observed at ca. 35% strain may be the initially generated network, which was not resolved at 0% strain. However, a similar analysis by Pei et al. ([18], Figure 5) indicates that the newly formed crack pattern is overshadowed by the pristine.

DLC coated samples show a slightly higher elastic modulus, tensile strength, and elongation at break. Keeping inaccuracies and scattering in the measurement in mind, it can be stated that the found differences are negligible. It can be speculated that the early fragmentation of the DLC coating (Figure 5b) leads to no great differences in mechanical properties.

As seen in the results section, the structured uncoated elastomers show significantly higher abrasion than the unstructured sample. This can be explained by the existing predetermined breaking points. This internal load results in a quicker material chipping behavior. The application of a nanostructuring could be a future improvement step. As expected, a DLC coating—regardless of the structuring—leads to a lower CoF and low wear. There are three explanations for this. First, the hysteretic friction of an elastomer–steel contact disappears with DLC coatings. Second, the bulging of the DLC layer—caused by the coating process—reduces the contact area and adhesive friction. Thirdly, the adaptability of a harder and stiffer DLC layer to the tribological partner is significantly lower than that of a softer elastomer. A closer look at the individual DLC coated samples leads to the following conclusions: The DLC layer on the unstructured elastomer specimen shows a cauliflower like “wavy cap” structure. The average waviness (approx. 80 µm) of this coating is higher than the average waviness of the layers on the structured specimens during tribological loading, and the protruding caps are crushed or worn and then lead to smaller fragmented still fixed coating pieces, as seen in the tensile tests. Fracture is assisted by the underlying shear stress introduced into the interfacial area by the coating. The initial small contact area at the top of the cap increases with ongoing fracture of the caps. This development is typically fast. The point structure S3 with DLC coating exhibits a similar behavior to the unstructured reference. Structure S3 does not strongly contribute to the reduction in the stress induced by the coating process as compared to the tile structures S1 and S2. The tile structures allow stress release during the coating process and the result is a more resilient surface structure. In the case of continuous loading, less heterogeneous damage is produced. The ability of the tiles to reduce shear strain during dynamic friction mitigates overloading effects. The CoF changes more steadily, and the structures do not (yet) show the larger scale fragmentation. In this case, the development of the coating change or destruction can be observed via the CoF. The findings for short term measurements in the way of initial damage of any cap structure and the resilience of tile structures to frictional shear load can be projected to long term behavior. The optimal limit would be found for a coated structure that only decays by DLC wear until the coating is sacrificed.

In this work, the chemical modification of the substrate induced by the laser irradiation was not taken into consideration. If present, we expect the modification to be limited to the trenches and a narrow region, directly adjacent to the trenches or dimples. The extent of a possible modification beyond the trenches would be dependent on the modification threshold and the spatial laser beam shape. The chemically modified surface could either promote or deteriorate the adhesion of the DLC. Therefore, the reported results could in part be influenced by a photochemical modification. However, as already elaborated, the effect would be limited to the trenches/dimples and their direct surrounding. Hence, if present, the effect may not greatly influence the overall adhesion. Moreover, the subsequent plasma treatment could remove the modified surface or largely overshadow any modification by laser irradiation.

## 5. Conclusions and Outlook

Elastomer substrates were laser microstructured with either two different tile patterns or a matrix of dimples and subsequently coated with DLC. Unstructured, DLC coated NBR samples showed a primordial crack network after coating. Under tensile stress, a further fragmentation of the DLC coating was observed, whereas a tile patterned substrate efficiently suppressed any further cracking except in the predisposed trenches. The coated samples showed a significantly reduced CoF compared to the uncoated ones. The structuring helped in the coated as well as in the uncoated state to lower the CoF and reduce wear. This effect partly wore off over time and was dependent on the surface structure. Macroscopically, the tile patterned samples (S1 and S2) look almost unaffected after tribology tests. We associate this finding with the superior resilience of the tile patterned coating to ongoing fragmentation. The stability of the positive effect of the CoF correlates with the stability of the structures against strain.

In future research, the possible chemical modification of the elastomer substrate induced by laser irradiation should be subject to further research. Alternatively, the structure could be transferred from a tool to the elastomer, excluding any contribution of possible chemical modifications. In addition, embossing would be the method of choice for commercial high volume production.

The positive effects of the surface structuring on the CoF, wear, and especially the formation of different crack patterns open up a wide range of new developments to control the surface structure of DLC coated rubbers and to enhance the tribological performance of, for example, rubber sealings.

## Figures and Tables

**Figure 1 materials-15-03271-f001:**
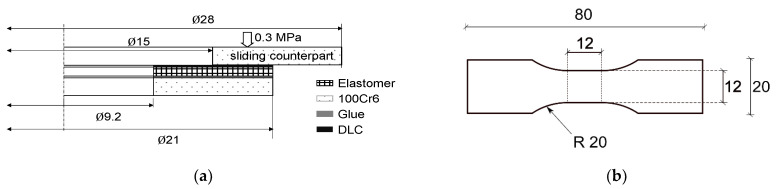
(**a**) Friction pair consisting of a test sample (DLC coated elastomer or elastomer) glued onto a 100Cr6 steel ring. A second steel ring acts as sliding counterpart. Only one half of the cross section is shown. (**b**) Dimensions of the tensile test specimens.

**Figure 2 materials-15-03271-f002:**
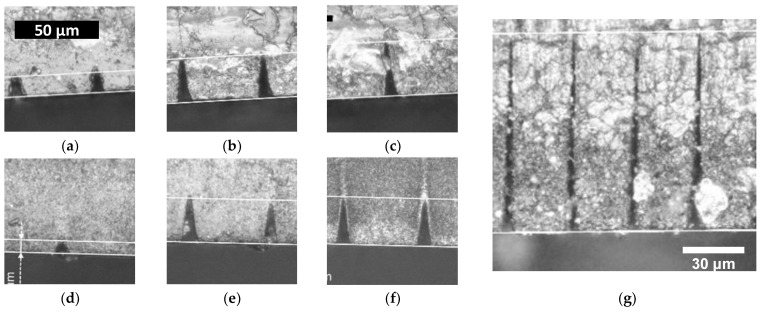
Comparison of SEBS G1650 (**a**–**c**) and NBR H-AT-68 (**d**–**g**) for 0.45 µJ pulse energy. The number of repetitions is increased from left to right from 5 over 15 to 20 and 70 for (**g**).

**Figure 3 materials-15-03271-f003:**
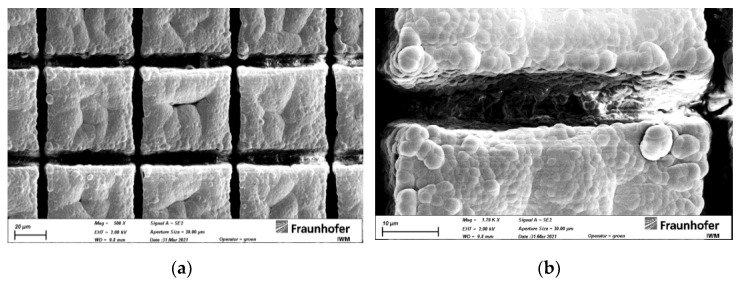
DLC coating on elastomer substrate with a hatch distance of 30 µm (S30). (**a**) overview with ×500 magnification and (**b**) detailed view into a trench (×1790).

**Figure 4 materials-15-03271-f004:**
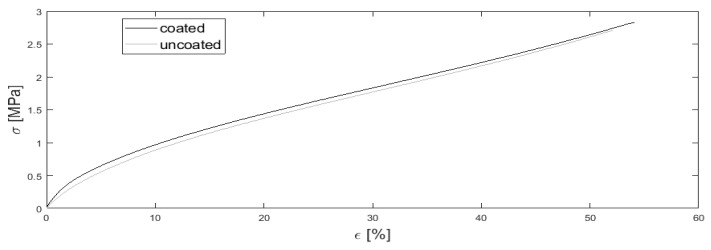
Stress (ε)–strain (σ) diagram of uncoated and DLC coated FKM samples.

**Figure 5 materials-15-03271-f005:**
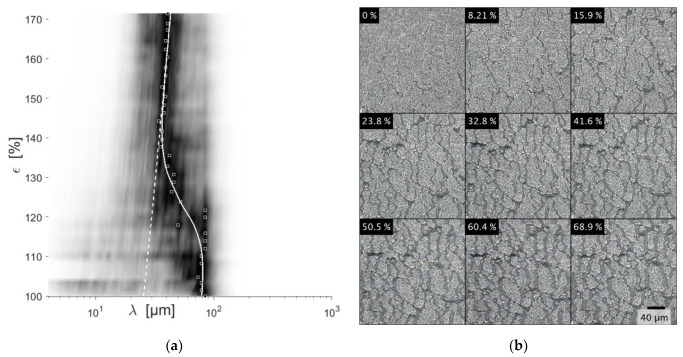
DLC coating on an unstructured substrate, showing the same section at different strains (**b**) and the corresponding SPD (**a**). The dotted white line highlights the measured strain. The solid white line is a logistic curve fitted to the maximum SPD values at each strain. It asymptotically approaches the measured strain. The SPD is normalized, and the contrast is enhanced by taking the fourth power. The maximum SPD value is marked by the squares.

**Figure 6 materials-15-03271-f006:**
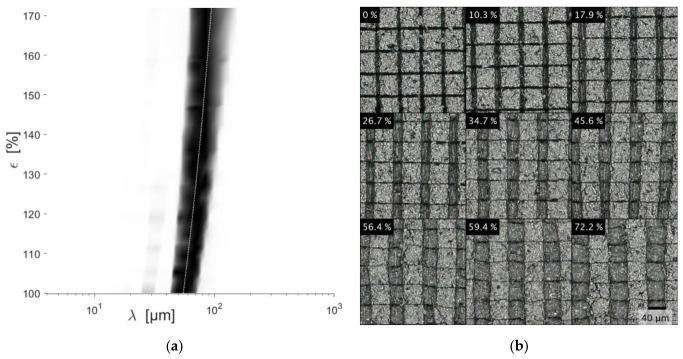
Microscope images (real space) of strained DLC coating with structure 1 (**b**) and the corresponding SPD (**a**). The dotted white line highlights the measured strain. The SPD is normalized, and the contrast is enhanced by taking the fourth power.

**Figure 7 materials-15-03271-f007:**
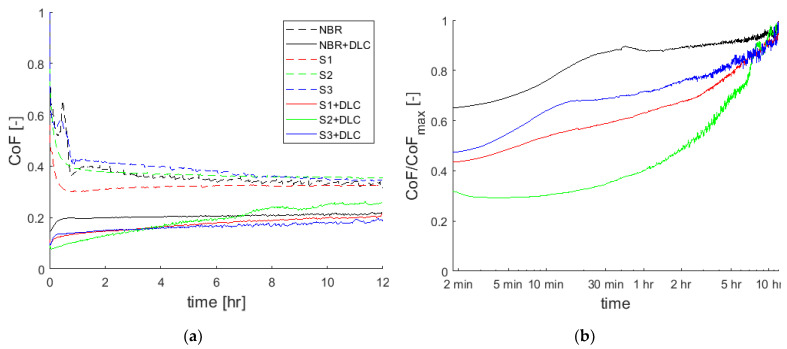
(**a**) Friction coefficient of different elastomer samples; (**b**) evolution of the normalized friction coefficient for DLC coated elastomers.

**Figure 8 materials-15-03271-f008:**
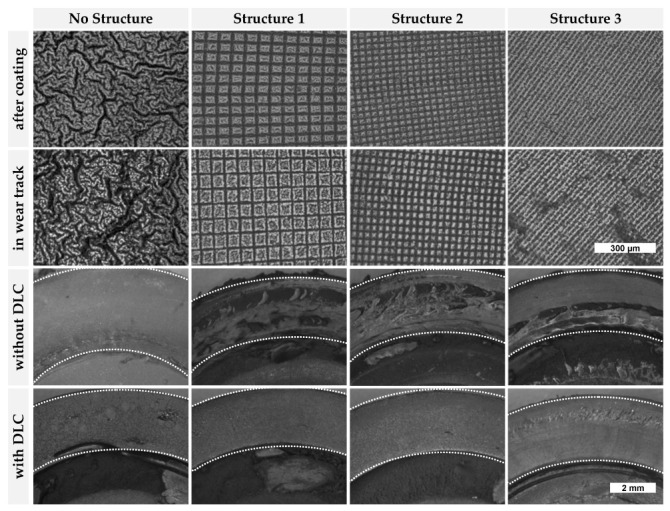
DLC coated surfaces before and after tribological loading.

**Table 1 materials-15-03271-t001:** Material characterization and threshold fluences F_Th_ used in this work. * Not applicable.

Rubber	Manufacturer	Hardness [°ShA]	Visual Appearance	Oil [phr]	Soot[phr]	F_Th_ [Jcm^−2^]
FKM 803915.3	Freudenberg	75	black	6.5	40.3	0.79
FKM 803927.4	Freudenberg	64	black	21.5	64.8	0.75
FKM Viton 75	DuPont	75	black	18.1	35.6	0.73
NBR 802607.2	Freudenberg	76	black	40	95.1	1.53
NBR 804506.1	Freudenberg	70	black	40.6	89.7	1.82
NBR H-AT-68-00	Goorex	70	black	-	-	0.82
SEBS G1650	Freudenberg	70	translucent	× *	× *	0.89
SEBS H1062	Freudenberg	77	translucent	× *	× *	0.99

## Data Availability

Data are available upon request.

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
