# Peer review of "Laser Structuring and DLC Coating of Elastomers for High Performance Applications"

_materials, 2022, doi:10.3390/ma15093271_

Round 1
Reviewer 1 Report
The manuscript describes the tribological properties and surface degradation of selected rubber seals modified after a LASER treatment and covered with DLC films by PE-CVD. In my opinion, before considering this manuscript eligible for publication, it should be reviewed in some aspect. One aspect was that the authors have not mentioned the physical and chemical modifications of different rubber to the LASER treatment. Is roughness increased? Reactive groups are formed on the surface?
Then, what is the structure of the DLC? What are its characteristics? How does it adhere to the rubber?
Is the thickness of DLC “film” of 2.5 µm? How was it estimated?
Moreover, I found some mistake in the text:
Line 17. What does it mean “Error! Reference source not found”?
Line 40: … of the DLC film is has been….
Line 98: CCP PE-CVD stays for capacitively-coupled plasmas (CCP) Plasma-enhanced Chemical vapour deposition (PE-CVD).
Line 187: DLC.
Line 261: colon followed by capital letter.
Line 266; the sentence is not clear.
Line 287: the CoF and.
Reviewer 2 Report
General comments:
- The topic is interesting and up-to-date. However, too little attention has been devoted to works on laser texturing of polymers. Research on elastomers has also been published in recent years. The Introduction section should be supplemented with relevant publications on laser texturing of polymers / elastomers.
- The work completely ignores the aspect of the chemical modification of the surface of rubber following laser treatment.
- The characteristic of the rubber samples is too sketchy. It does not contain any details on sample composition.
- The description of some experimental techniques is missing in the Experimental section.
Specific comments:
- The sentence (line 17-19): “In this work a nature inspired approach (Error! Reference 17 source not found.Figure 1b) is applied to render the stiff coating flexible and resilient to delamination at the same time by direct patterning” is not clear.
What is presented on Figure 1b, snake’s skin? What are the similarities between a floe and a snake's skin?
- Line 35-36: “A hard, protecting, and low friction coating such as diamond like carbon (DLC) would significantly enhance the performance of dynamic rubber seals”.
This sentence is true, but requires a more in-depth explanation of how a thin film may affect the dynamic loss of an elastomeric material.
- Line 40: “Especially the microstructure of the DLC film is has been attributed to adhesion and flexibility performance [6–9]”. “is” has to be deleted.
- Materials studied, listed in Table 1, should be better characterized. The kind of filler and the filler loading are likely to play an important role in the modification.
- Line 100-101: “… coating with tetramethylsilane (TMS, 17 SCCM) as precursor gas. The main coating gas (toluene) …”.
Neither TMS nor toluene are gases. The Authors used too much of a mental shortcut.
- Section 2.4
Why only FKM samples were subjected to quasistatic tensile tests? What is “transition radius”? The description of the analysis of the results of mechanical tests (SPD), conducted in a non-standard manner, should be deepened.
- Section 2.5
A schematic drawing of the geometry of the friction pair would be useful.
- What is presented on Figure 2g? There is no caption for the photo g. The description of microscopic investigations is missing in the Experimental section.
- Line 138-139: “Both show a similar dependency on the number of scans as indicated by the similar single shot threshold fluence (Table 1)”. Table 1 doesn’t contain data the Authors referred to.
- Line 144-145
“The DLC coating had a thickness of 2.5 µm …”. Where was the thickness of the coating measured: on the outer surface, in the depressions or on the walls of the depressions? Do not the Authors think it should each time be different? Should not the DLC thickness be dependent on substrate pattern geometry?
Additionally, the description of microhardness measurements is missing in the Experimental section.
- Line 149-151: “The elastic modulus of the uncoated sample (19.1 MPa) is lower than the elastic modulus of the DLC-coated sample (28.9 MPa).
What do the Authors mean by the modulus of the DLC-coated sample?
Round 2
Reviewer 1 Report
After the second revision, the quality of the manuscript was improved, but I still find a lack the absence of the characterization of the chemical composition of the DLC. It seems like the authors consider the DLC to exist in one defined compound. I suggest them to check the quality of the coating by Raman Sprectroscopy or X-ray photoelectron spectroscopy (XPS)/ Auger electron Spectroscopy (AES).
Lines 107-109 The authors have to check the punctuation;
Line 199 the capital letter A after the comma;
Line 327 I suppose that “elastic modus” must be changed in “elastic modules”.
Reviewer 2 Report
Dear Authors, please find comments to the points addressed in the first review:
- Introduction: Chemical modification has been confirmed for different polymers for USP or nanosecond pulse length regimes [25–34].
The cited references have to be discussed separately in more detail.
- Chemical modification is likely to take place not only inside the trenches. The chemical effect is likely to overlap or at least to stress relief produced by geometrical effect. Could the Authors cite a literature claiming that the modification is limited to a narrow range close to the trenches and inside the trenches? Why do the Authors think that chemical modification induced by the laser radiation might negatively influence the adhesion between substrate and film?
3 and 8. The composition of the samples studied, especially the kind and a filler content is very important from the point of view of the effect of laser treatment, especially from the point of view of the accuracy of the shape and depth of the trenches.
- I accept the explanation of the principles of SPD technique, but I have no idea why Appendix A has a picture of the friction pair components, which I think fits better in the chapter 2.5.
- Have the pictures been left intentionally?
- This is exactly what was expected. Influence of the reduction of adhesional component on the development of the hysteretical component of friction.
- I suggest to place a scheme presenting the dimensions of the test specimen.
- The schematic drawing, presenting the geometry of the friction pair has not been supplied. The pictures presented in the Appendix is not unequivocal.
- ISO 864 has nothing to do with microhardness measurements.
- The Authors have not answered my question. I have not noticed any changes in the text, mentioned by the Authors.
